# Hyper Cross-Linked Polymers as Additives for Preventing Aging of PIM-1 Membranes

**DOI:** 10.3390/membranes11070463

**Published:** 2021-06-23

**Authors:** Federico Begni, Elsa Lasseuguette, Geo Paul, Chiara Bisio, Leonardo Marchese, Giorgio Gatti, Maria-Chiara Ferrari

**Affiliations:** 1Dipartimento di Scienze e Innovazione Tecnologica, Università degli Studi del Piemonte Orientale “Amedeo Avogadro”, Viale Teresa Michel 11, 15121 Alessandria, Italy; federico.begni@uniupo.it (F.B.); geo.paul@uniupo.it (G.P.); chiara.bisio@uniupo.it (C.B.); leonardo.marchese@uniupo.it (L.M.); 2School of Engineering, University of Edinburgh, Robert Stevenson Road, Edinburgh EH9 3FB, UK; e.lasseuguette@ed.ac.uk; 3CNR-SCITEC Instituto di Scienze e Tecnologie Chimiche “G. Natta”, Via C. Golgi 19, 20133 Milano, Italy

**Keywords:** mixed-matrix membranes, physical aging, hyper cross-linked polymer, gas permeation, ^13^C spin-lattice relaxation times, SS-NMR spectroscopy

## Abstract

Mixed-matrix membranes (MMMs) are membranes that are composed of polymers embedded with inorganic particles. By combining the polymers with the inorganic fillers, improvements can be made to the permeability compared to the pure polymer membranes due to new pathways for gas transport. However, the fillers, such as hyper cross-linked polymers (HCP), can also help to reduce the physical aging of the MMMs composed of a glassy polymer matrix. Here we report the synthesis of two novel HCP fillers, based on the Friedel–Crafts reaction between a tetraphenyl methane monomer and a bromomethyl benzene monomer. According to the temperature and the solvent used during the reaction (dichloromethane (DCM) or dichloroethane (DCE)), two different particle sizes have been obtained, 498 nm with DCM and 120 nm with DCE. The change in the reaction process also induces a change in the surface area and pore volumes. Several MMMs have been developed with PIM-1 as matrix and HCPs as fillers at 3% and 10wt % loading. Their permeation performances have been studied over the course of two years in order to explore physical aging effects over time. Without filler, PIM-1 exhibits the classical aging behavior of polymers of intrinsic microporosity, namely, a progressive decline in gas permeation, up to 90% for CO_2_ permeability. On the contrary, with HCPs, the physical aging at longer terms in PIM-1 is moderated with a decrease of 60% for CO_2_ permeability. ^13^C spin-lattice relaxation times (T1) indicates that this slowdown is related to the interactions between HCPs and PIM-1.

## 1. Introduction

Membrane-based materials have played an important role in the field of gas separation [1]. The relatively low energy requirement of membrane processes [2,3] makes them an attractive alternative to more intensive separation processes, such as pressure swing adsorption, chemical absorption and cryogenic [4,5,6,7]. A promising area of development for gas separation is the use of hybrid membranes or mixed-matrix membranes (MMMs). MMMs are composite membranes made by combining a filler (dispersed phase) and a polymer matrix (the continuous phase). By using two materials with different transport properties, these membranes have the potential to synergistically combine the easy processability of polymers and the superior gas-separation performance of filler materials, and therefore provide separation properties surpassing the Robeson upper bound [8]. In particular, enhancements have been noticed with nanostructured and highly porous additives, such as metal–organic frameworks, zeolites, carbon nanotubes or hyper crosslinked polymers (HCP) [9,10,11,12,13,14,15]. The addition of porous fillers in the continuous phase plays an important role in the transport properties of MMMs, providing new pathways for gas transport, hence increasing gas permeability [16,17,18,19,20,21]. Furthermore, the incorporation of fillers might be a solution to tackle the physical aging of glassy polymers [22,23,24,25,26,27].

HCPs are a class of high surface area amorphous polymers, which can be obtained either from the post-crosslinking of polystyrene-type precursors in their swollen state, or from the cross-linking of small building blocks, following Friedel–Crafts chemistry [28,29]. The resulting polymers consist of aromatic rings joined together through aliphatic bridges. The formation of a highly interconnected network results in a stable nanoporous structure. HCPs present several advantages, they are stable, tunable, inexpensive, scalable [30] and present high surface areas (1200–2000 m^2^g^−1^) [29], which allow high gas uptakes [31]. For these reasons HCPs are attractive porous solids to be used as fillers within mixed-matrix membranes in order to improve separation performances. Numerous studies show their ability to provide high permeability membranes which can surpass the classical Robeson upper bound. Hou et al. [32] developed MMMs with Poly(1-trimethylsilyl-1-propyne) (PTMSP) matrix and 10 wt % HCP as fillers, based on α,α′-dichloro-p-xylene. Considerable selectivity improvement was achieved thanks to the efficient nanoparticle dispersion and sufficient interaction between the matrix and the filler. The enhancement of gas permeability of MMMs was attributed to the pore channels added by the highly crosslinked fillers which provided additional pathways for gas transport. Lau et al. [33] studied the physical aging of these MMMs. Over time, the CO_2_/N_2_ separation performance increased whilst CO_2_ permeability remained stable. This selective-aging effect is explained by the retention of fractional free volume (FFV) content by HCP particles. The ^13^C solid-state NMR spectroscopy showed that the bulky part of the PTMSP chains was immobilized by the HCP. Mitra et al. [34] developed MMMs composed of PIM1 and HCP based on poly(vinyl benzyl chloride). The incorporation of fillers induced an increase of gas permeability due to the higher Brunauer–Emmett–Teller (BET) surface area of HCP (1700 m^2^g^−1^) compared to PIM1 (750 m^2^g^−1^). On the contrary, the CO_2_/N_2_ selectivity decreased with the presence of HCP, especially with high amounts of filler (>15 wt %). Moreover, the presence of HCP retarded the physical aging of the MMM with a lower decrease in CO_2_ permeability as compared to PIM-1 alone.

In this paper, we report on the synthesis of new HCPs based on the Friedel–Crafts reaction between a tetraphenyl methane monomer and a bromomethyl benzene monomer. Depending on the solvent used during the reaction, i.e., dichloroethane (DCE) or dichloromethane (DCM), HCPs with different particle sizes were obtained. To the best of our knowledge, the effect of the synthetic conditions on the HCP particle sizes has not been covered in the scientific literature. The synthesized materials have been characterized in terms of their structure and properties. The effect on the membrane gas separation performance over time was investigated when HCPs are used as additives in PIM-1 membranes.

## 2. Materials and Methods

### 2.1. Materials Preparation

The synthesis of PIM-1 was performed following the procedure reported in literature [35], by mixing anhydrous K_2_CO_3_ (11.05 g, 80 mmol), 5,5′,6,6′-tetrahydroxy- 3,3,3′,3′-tetramethyl-1,1′-spirobisindane (3.4 g, 10 mmol), and 2,3,5,6-tetrafluoroterephthalonitrile (2.0 g, 10 mmol) in anhydrous dimethylformamide (65 mL) at 338 K for 72 h under an N_2_ atmosphere. On cooling, the mixture was added to water (500 mL), and the crude product collected by filtration. Repeated precipitations from methanol gave 4.96 g (92% yield) of fluorescent yellow polymer (PIM-1) with a Mw ~ 87,000 g mol^−1^. 3,3,3′,3′-Tetramethyl-1,1′-spirobiindane-5,5′,6,6′-tetraol and 2,3,5,6- tetrafluorophthalonitrile were purified before use by recrystallization in methanol and ethanol, respectively. Ethanol and methanol were purchased from Fisher Chemicals. Anhydrous dimethylformamide (99.9%) and 5,5′,6,6′-tetrahydroxy- 3,3,3′,3′-tetramethyl-1,1′-spirobisindane (97%) were purchased from Alfa Aesar. Potassium carbonate (99.5%) and 2,3,5,6-tetrafluoroterephthalonitrile (98%) were purchased from Fluorochem.

HCPs were named ABT01 and ABT02, AB from aluminum bromide and T from tetraphenyl methane, while 01 is used for the material obtained by using dichloromethane in the synthetic procedure and 02 for the material obtained using dichloroethane.

Here the synthesis of ABT01 is reported, however the same procedure was employed for the material named ABT02, except that dichloroethane and a temperature of 80 °C were used instead of dichloromethane at 35 °C.

ABT01 synthetic procedure was carried out in a 250 mL three-necked bottom flask by adding 1 g (3.12·10^−3^ mol) of tetraphenyl methane (TPM, Capot Chemical Company (97%)) to 90 mL of dichloromethane (DCM) (Sigma-Aldrich ≥ 99,8%) at room temperature. After approximately 10 min the addition of 10.02 g (2.8 × 10^−2^ mol) of the cross-linker 1,3,5-tris(bromomethyl)benzene (Sigma-Aldrich, >97%) was carried out which after 10 min, which was followed by the addition of 7.46 g (2.8 × 10^−2^ mol) of the catalyst aluminum (III) bromide (Sigma-Aldrich, ≥98%). The reaction mixture was left under stirring for approximately 20 min and then heated under reflux over night at 308 K. A molar ratio of 1:9:9 between the monomer, the cross-linker and the catalyst was used for the synthesis of both materials. After approximately 22 h the resulting mixture appeared as a dark brown gel. The reaction was quenched by addition of diluted HCl with deionized water (*v*/*v* 2:1), then the material was washed with ethanol and deionized water. After the washing step, the material was put in an oven at 343 K for 24 h. ABT01 appears as a brown powder.

The reaction scheme associated with the synthesis of ABT materials is reported in Figure 1.

### 2.2. Membranes Preparation

Solution casting [35] at ambient conditions was used to fabricate dense film membranes with a filler content of 3 wt % and 10 wt % (with respect to PIM-1 weight). For the preparation of the membranes a suspension of filler (6 mg for 3 wt % or 20 mg for 10 wt %) in 5 mL of CHCl_3_ was sonicated with an ultrasound probe (Fisher Scientific, Model CL18, 120 W) for 1 h by using a water bath to maintain the flask at room temperature. Meanwhile, 200 mg of PIM-1 was dissolved in 5 mL of CHCl_3_. After complete dissolution, the PIM-1 solution was added to the additive suspension with other 5 mL of CHCl_3_. The mixture was then sonicated again for 2 h at room temperature. The resulting solution was poured into a 5 cm glass Petri dish. The membrane was allowed to form by slow solvent evaporation for 24−36 h under a fume cupboard. Five membranes were obtained, namely, a pure PIM-1 membrane (PIM-1), two MMMs composed of PIM-1 and ABT01 as a filler at 3 and 10 wt % named, respectively, PIM1-ABT01-3% and PIM1-ABT01-10% and two MMMs composed of PIM-1 and ABT02 as a filler at 3 and 10 wt %, namely, PIM1-ABT02-3% and PIM1-ABT02-10%. After the drying steps, the thickness of the membranes was determined with a digital micrometer (Mitutoyo). Before performing permeability measurements, the membranes were immersed in methanol for 2 h, followed by a drying step under a fume cupboard for 1 h and then under vacuum at room temperature overnight. It is well known that the permeability performances of PIM-1 membranes may vary based on the casting conditions (for example, the choice of solvents) and the history of the sample. Alcohol washes away residual casting solvent and provides a comparable starting point for evaluation of different membranes [35].

### 2.3. Membranes Characterizations

Permeation measurements—The permeation properties of the MMMs were tested using the constant volume-variable pressure method in an in-house built time-lag apparatus of which a schematic can be seen in Figure 2.

The permeation cell consists of three main parts, namely, upstream, permeation cell and downstream. The upstream, or feed side, of the permeation cell consists of a controlling valve, a pressure gauge and a 2000 cm^3^ volume gas reservoir. The sample is positioned in the gas permeation cell and sealed with two rubber O-rings. The downstream volume is fixed and a pressure transducer is used to detect pressure changes.

The permeability is obtained from the evolution of pressure of the downstream side (Brooks Transducer 1000 mBar, CMC Series). The permeability coefficient, *P*, was determined from the slope of the pressure vs. time curve under steady state condition (Equation (1)).
(1)$$P\left(G\right)=\frac{l}{A}\frac{V_{down}\;}{P_{up}RT}{\left(\frac{dP_{down}}{dt}\right)}_{ss}$$
where *l* is the membrane thickness, *A* is the membrane area, *V_down_* is the downstream volume, *P_up_* is the upstream pressure (*P_up_* = 1.1 bar), *P_down_* is the downstream pressure, *T* is the temperature recorded during analysis and *R* is the gas constant.

Before each experiment, the apparatus is vacuum-degassed and a leak rate is determined from the pressure increase in the downstream part.

The ideal selectivity between two gas species *i* and *j* is the ratio of the two single-gas permeabilities (Equation (2)).
(2)$$\alpha_{ij}=\frac{P\left(i\right)}{P\left(j\right)}$$

For the aging tests, the initial measurement is performed right after the methanol treatment. The membrane is then stored in a sealed plastic bag at ambient temperature and tested over time.

Scanning Electron Microscopy (SEM)—The morphological properties of the membranes were examined with a Quanta 200 FEI (Hillsboro, Oregon) operating at 20 kV and equipped with an EDAX (Mahwah, New Jersey) EDS attachment. Before SEM analysis, the samples were prepared by sputtering with a 20-nm layer of gold to form a conductive surface.

N_2_ physisorption analysis—The specific surface area (SSA) was measured by means of nitrogen adsorption at liquid nitrogen temperature (77 K) in the pressure range of 1 × 10^−6^ Torr to 1 P/P_0_ by using an Autosorb-1-MP (Quantachrome Instruments). Prior to adsorption, the samples were outgassed for 3 h at 423 K, (final pressure lower than 10^−6^ Torr). The SSA of the samples was determined by the Brunauer−Emmett−Teller (BET) equation, in a pressure range 0.05−0.15 P/P_0_ selected to maximize the correlation coefficient of the fitted linear equation. The pore size distribution was calculated by means of the nonlocal density functional theory (NLDFT) method for slit pores.

FT-IR spectroscopy—Infrared spectra were collected on a Thermo Electron Corporation FT Nicolet 5700 spectrometer (resolution 4 cm^−1^). Pellets were prepared by mixing the prepared materials with KBr (1:10 weight ratio) and placed into an IR cell with KBr windows permanently connected to a vacuum line (residual pressure: 1 × 10^−4^ mbar), allowing all treatments to be performed in situ. Samples were degassed for 3 h, using an oil-free apparatus and grease-free vacuum line.

SS-NMR spectroscopy—Solid-state NMR spectra were acquired on a Bruker Avance III 500 spectrometer and a wide bore 11.7 Tesla magnet with operational frequencies for ^1^H and ^13^C of 500.13 and 125.77 MHz, respectively. A 4 mm triple resonance probe, in double resonance mode, with magic angle spinning (MAS) was employed in all the experiments. The as-cast polymer membranes were cut into small pieces so that they could be packed in a 4 mm Zirconia rotor and were spun at a MAS rate of 12 kHz. For the ^13^C cross-polarization (CP) MAS experiments, the proton radio frequencies (RF) of 55 and 28 kHz were used for initial excitation and decoupling, respectively. During the CP period the ^1^H RF field was ramped using 100 increments, whereas the ^13^C RF field was maintained at a constant level. During the acquisition, the protons were decoupled from the carbons by using a Spinal-64 decoupling scheme. A moderate ramped RF field of 55 kHz was used for spin locking, while the carbon RF field was matched to obtain optimal signal (40 kHz). T1 measurements were performed with a CPXT1 pulse sequence using a 10 ms spin-lock of 55 kHz and 40 kHz for ^1^H and ^13^C, respectively, immediately followed by π/2 − τ − π/2 sequence on ^13^C with variable delay (τ) ranging from 0.1 to 45 s. Spectra were recorded with a spectral width of 42 kHz and 256 transients were accumulated at 298 K. A line broadening of 50 Hz and zero filling to 2048 points were used. All chemical shifts are reported using δ scale and are externally referenced to TMS at 0 ppm. Data analyses were performed using Bruker software Dynamics Center, version 2.5.6 and T1 curves were obtained by plotting the intensity of the carbon signals versus time. A single exponential decay was used to fit the data using the following equation (Equation (3)):(3)It=I0e−tT1

## 3. Results

### 3.1. Characterization of the HCPs

#### 3.1.1. SEM

In Figure 3, SEM images of ABT materials are reported.

ABT01 and ABT02 appear as aggregates of round-shaped particles. Particle dimensions for ABT01 are 498 ± 39 nm while for ABT02 they are 120 ± 23 nm. From the Energy Dispersive X-Ray elemental analysis (EDX) reported in Appendix A, it is seen that ABT materials are mainly composed of carbon, which is expected. The presence of aluminum and bromine is to be attributed to unreacted catalyst species, which is present in higher quantities in ABT01. Bromine could also be present within the polymeric framework as partially unreacted crosslinker species as well as side reaction products.

#### 3.1.2. N_2_ Physisorption Analysis

The results of the N_2_ physisorption analysis are reported in Figure 4.

As it can be seen in Figure 4, ABT isotherms (with relative upper pressure of only 0.76–0.77) do not show a horizontal plateau after the initial filling of the micropores, therefore a clear classification of the isotherms cannot be made. In particular, a distinction between type I and II is difficult. A fair amount of gas (>200 cm^3^/g) is adsorbed at low relative pressures (up to 0.1 P/P_0_), indicating permanent micro porosity for both materials. Towards higher values of P/P_0_ a gradual increase in the amount of adsorbed nitrogen is observed indicating the filling of mesopores in the range between 0.45 and 0.8 P/P_0_. Open hysteresis loops for both materials are observed for the whole desorption branch, which is consistent with swelling effects of the polymeric network, due to gas sorption [36,37]. The non-reversible desorption at low relative pressures indicates that N_2_ could either be trapped in pockets or free volume elements with a size comparable to that of the N_2_ molecule, or swelling of ABT is locking some of the pockets or free volume elements on the time scale of the experiments, or in combination. From the data reported in Table 1 it can be seen that ABT02 possess higher surface area with respect to ABT01, namely, 990 m^2^/g versus 823 m^2^/g; in addition, pore volume values associated with both micro- and mesoporosities are slightly higher in ABT02, which is probably a solvent-induced effect since dichloroethane is known to be a particularly suited solvent for the development of high porosity degree in hyper crosslinked polymers [38].

#### 3.1.3. FT-IR Spectroscopy

IR spectra of ABT materials are reported in Figure 5 with the assignments of the IR absorption bands in Table 2.

The IR spectra of ABT materials present signals in the ranges 3200–2800 cm^−1^ and 1750–400 cm^−1^. In the high wavenumber region, broad bands are found centered at 3053, 3020, 2965, 2922, 2870 and 2845 cm^−1^. The signals at 3053 and 3020 cm^−1^ are respectively assigned to the asymmetric and symmetric stretching of aromatic C-H groups [39,40]. Between 3000 and 2800 cm^−1^, the signals of aliphatic C-H stretching vibrations are found. In particular, the band at 2965 cm^−1^ is assigned to the C-H asymmetric stretching mode of methyl groups [37,39,40] while the band at 2922 cm^−1^ is assigned to the asymmetric stretching mode of the -CH_2_- group [39,40]. The corresponding symmetric stretching vibrations are found at 2870 cm^−1^ for the methyl group and at 2845 cm^−1^ for the methylene group [37,39,40]. The ratio between the intensity of the aliphatic over aromatic C-H stretching modes bands is higher for the ABT02 with respect to ABT01, which could be an indication of a more crosslinked network. In the low frequency region, a series of signals is found, and assignments have been made for the main absorption bands, namely, the bands at 1698, 1601, 1507, 1446, 1270, 1020, 898 and 705 cm^−1^. Between 1700 and approximately 1200 cm^−1^, signals associated with stretching modes of -C=C- bonds are found [39,40]. The band at 1698 cm^−1^ is associated with hindered vibrations of the aromatic rings, probably due to high crosslinking degree of the polymeric framework [41]. The relative intensity of this signal slightly increases for ABT02 with respect to ABT01. This can be linked to higher interconnectivity of the polymeric network of ABT02, probably as a consequence of the higher temperature adopted during the synthetic procedure which accelerated the crosslinking reaction. Additional evidence of the higher crosslinking degree for ABT02 is also the lower intensity of the signal at 1207 cm^−1^ assigned to skeletal C-C bond vibrations [39,40]. The absence of a sharp peak around 1500 cm^−1^ and the presence of the intense signal at 1601 cm^−1^ are indications of possible predominant meta substitution of the aromatic rings, which is expected when AlBr_3_ is used in the synthetic procedure [39,40,42].

In-plane bending modes of aromatic C-H groups are found between 1225 and 950 cm^−1^ [40] while out-of-plane bending modes are found between 1020 and 700 cm^−1^ [40]. The presence of the intense signal centered at 898 cm^−1^ may be interpreted as a sign of 1,3 substitution of the aromatic ring [40].

Between 700 and 600 cm^−1^ the stretching vibrations of the C-Br bond are found. For ABT01, a weak signal at 636 cm^−1^ is found, which indicates that a small amount of bromine is directly linked to the polymeric network, probably as –CH_2_Br groups, since the aromatic C-Br bond stretching vibrations are found around 680 cm^−1^ [40]. For ABT02, only very weak broad bands are observed in the region of the C-Br stretching modes.

#### 3.1.4. SS-NMR Spectroscopy

In Figure 6, ^13^C CPMAS NMR spectra of the two samples are reported.

Peaks associated with aromatic carbons are found in the region between 150 and 120 ppm. Around 142 ppm the signal is associated with both the carbon directly attached to the central quaternary carbon and the carbon directly attached to the methylene group of the crosslinker [43].

Aromatic carbons associated with C-H groups are found at 134 and 128 ppm [43]. The central quaternary carbon of the monomer unit is found at 58 ppm while around 35 ppm the carbons associated with the methylene group of the crosslinker are found [43]. In the region around 40–35 ppm the signals associated with -CH_2_Br can also be found [44]. Carbons associated with methyl groups are found around 17 and 12 ppm [37,43]. It is interesting to observe the presence of these functional groups since they are not present in either the monomer or crosslinker. This finding is also confirmed by the FT-IR analysis (vide supra). As a possible explanation, the presence of ethyl groups can be a consequence of a small fraction of dichloroethane molecules reacting with the catalyst while methyl groups could be explained via mechanisms involving the carbocationic sites of the crosslinker. This type of functional group was previously found in hyper crosslinked aromatic polymers and derives from secondary reactions that can occur in the presence of dichloroethane solvent [43].

### 3.2. Membranes Characterization

Membrane photographs of PIM-1 and PIM-1-ABT0 membranes are presented in Figure 7.

As Figure 7 shows, good dispersion of the filler within the membrane has been obtained. No aggregates were visible.

#### 3.2.1. Permeability Measurements

Freshly cast membranes

Table 3 shows the separation performances of PIM-1 and the four MMMs after MeOH treatment, at t_0_.

The permeation data of PIM-1 are in the range of those reported in the literature [35]. PIM-1 is highly permeable to CO_2_ and presents a reasonable CO_2_/N_2_ selectivity as well. By adding 3 wt % of the filler ABT02, a decrease of 35.2% in CO_2_ permeability with the CO_2_/N_2_ selectivity almost unaffected is observed. Addition of 10 wt % of ABT02 results in a drop of 44% in CO_2_ permeability while an increase in selectivity from 15 to 18 is also observed. The decline in permeability could have multiple sources and it might be due to densification of the PIM-1 polymeric chain, filling of fractional free volumes by the HCP particles or a partial blockage of the HCP pores by polymer chains [45]. Another possible explanation could be that non-covalent interactions between the fillers and the PIM-1 matrix do not allow for the complete regeneration of the MMMs via methanol treatment. For the MMM sample prepared with the addition of 10 wt % of ABT01, we have a similar separation performance decline of 34% in CO_2_ permeability, but also a drop of 13% in selectivity. This drop might be explained as for ABT02 with a possible rigidification of the polymeric chain due to the filler, by occupation of fractional free volume within the PIM-1 matrix by addition of the fillers or by a partial blockage of the ABT01 pores by PIM-1 chains. The addition of ABT01 at low content stands out as it induces an increase in CO_2_ permeability and selectivity. The addition of ABT01 with a 10 wt % loading causes a drop in the selectivity while the addition of ABT02 with the same loading causes the CO_2_/N_2_ selectivity to increase. This may be ascribed to particle dimension effects, meaning that the larger particles associated with ABT01 could partially disrupt chain packing in the PIM-1 matrix resulting in the formation of non-selective gas diffusion pathways.

Aging behavior—Permeability measurements were conducted over the course of approximately two years. Measurements were performed after leaving the sample under vacuum over-night at room temperature.

Figure 8 summarizes the aging characteristics of the pristine PIM-1 and MMMs between t_0_ and t_f_. The mixed matrix containing 3% of ABT01 was only measured up to 250 days.

In Figure 9 the initial drop in the normalized permeation data is reported as a function of time over approximatively the first year of aging.

All the samples present a drop in permeability with time. Aging effects on all samples are clearly visible in the first few days after methanol treatment regardless of the size of the particles (i.e., ABT01 or ABT02) or the concentration of the filler (i.e., 3 or 10 wt %). PIM-1 displays the classical aging behavior exhibited by polymers of intrinsic microporosity [46,47], namely, a progressive drop in the permeation capacity associated with both CO_2_ and N_2_ caused by the collapse of free volume [46]. After approximatively one year of aging, CO_2_ permeability of pure PIM-1 decreases by 85% with respect to t_0_ value, while for N_2_ it reduces by 90% of the initial permeability, hence an increase in selectivity over time towards CO_2_ is observed.

As shown in Figure 9, the nanocomposite membranes show a better resistance to the aging than the pristine membrane at 200 days where the permeability drop appears to have stopped. The addition of the ABT compounds leads to an arrest in polymer aging and permeability loss. The particles incorporated within the polymer prevent the collapse of the free volume [22,25,27,33,34]. The HCPs possess a rigid network due to the high level of crosslinks which prevents the collapse of the structure. The PIM1-ABT01-3% membrane shows a different behavior compared to the other samples, with a slower rate of aging more similar to PIM-1 with aging progressing even when the other samples have stabilized. In Table 4 the final permeability at ~850 days is reported for the mixed matrices that had stopped aging, confirming that the insertion of the HCPs has prevented further aging compared to the one-year mark and with no significant difference between the filler produced with a different solvent (i.e., of different particle sizes) or the concentration of the filler. Only 3% of the filler ABT02 is sufficient to inhibit aging compared to the pristine PIM-1.

#### 3.2.2. SS-NMR ^13^C T1 Measurements

The typical ^13^C CPMAS NMR spectra for PIM-1 and PIM1-ABT-based as-cast membranes are shown in Figure 10. The only visible difference between these spectra is that in the MMMs we see an additional broad peak due to aromatic carbons (highlighted by red arrow) belonging to the ABT fraction. The ^13^C resonances originating from the PIM-1 backbone have enough peak resolution and intensities such that the ^13^C spin-lattice relaxation time (T_1_) measurements have been carried out. The influence of additives on the relaxation behavior of individual carbons in PIM-1 can reveal the molecular level dynamics in the polymer membranes.

The T_1_ values are related to the mobility/rigidity of specific carbon atoms within the polymer and the ^13^C relaxation studies allow one to estimate the aging in polymer membranes. We have recorded the ^13^C T_1_ values, at ambient temperature, on the freshly cast (*t_0_*) as well as on the one-year aged (*t_12_*) membranes. The relative changes ((*t_12_ − t_0_*)/*t_0_*) in the ^13^C spin–lattice relaxation time values, which have been evaluated to estimate the aging behavior in PIM1 based membranes, are shown in Figure 10d.

As the aging starts, an increase in the T_1_ values for the polymer carbons is expected due to the rigidification of the polymer backbone and the gradual reduction in its excess free volume [22,25]. This is the case in the PIM-1 membrane where the backbone carbons (C1, C2, C3, C6 and C7) have longer T_1_ values at *t_12_*. Similarly, uniformly higher spin-lattice relaxation times at *t_12_* were detected for all carbons in the PIM1-ABT01-3% aged composite membrane. These results show that as the polymer membrane ages, the relaxation times of polymer carbons increases, and the chain mobility decreases due to densification. These results confirm the similar trend in permeability over time for the PIM1-ABT01-3% and pristine PIM-1 membranes.

As far as the other composite membranes (namely, PIM1-ABT01-10%, PIM1-ABT02-3%, and PIM1-ABT-02 10%) are concerned, significant reductions in the T_1_ values at *t_12_* for all carbons have been noted. In particular, carbons C1–C4 show the significant decrease in ^13^C T_1_ values between *t_0_* and *t_12_*, evidencing the preferable influence of additives on those aromatic carbons. These reductions in the spin-lattice relaxation times on the aged membranes are due to greater molecular mobility of the polymer backbone carbons because of their preferential interactions with ABT components. Previous studies [25,33] have shown that Porous Aromatic Frameworks (PAF) based additives have moderated the aging in PIM-1-based MMMs. Here ABT-based additives should make non-covalent interactions with the PIM-1 backbone, thus preventing the compactification of the polymer chains. On average, weak non–covalent interactions are expected to be at play between the filler and the PIM-1 matrix. Shorter T_1_ values could be observed, as is in the case of the present study, if these interactions could result in an enhancement of the mobility of chain backbones, which would also cause a freeing of some fractional free volume, in turn increasing the segmental motion further. As the excess free volume gradually increases, physical aging is diminished in the longer terms in the remaining three ABT-based composite membranes (PIM1-ABT01-10%, PIM1-ABT02-3%, and PIM1-ABT02-10%).

In summary, although the carbon atoms in PIM-1 can be categorized into flexible and bendable units or bulky units or contortion points, their effective packing can lead to diminution of chain motions. On the contrary, when a PIM-1 membrane is cast with the addition of ABT as a filler, new pathways for the gas transport could be generated. In addition, the intimate mixing at the molecular level introduces non-covalent interactions which lead to the retention of fractional free volume in the MMMs. Since the relaxation behavior of aromatic carbons in PIM-1 are mostly influenced, π-π stacking-based interactions can be envisaged as the ABT additive belongs to a hyper-crosslinked aromatic polymer system.

The difference in aging behavior between ABT01 and ABT02 MMMs could be ascribed in part to the difference in textural properties between the two fillers, however the difference in particles size between the two fillers could lead to higher contact surface for ABT02, being the fillers with the smaller particles (see SEM characterization).

The interactions induce anti-aging properties into PIM-1 membranes revealing the chemical compatibility between the ABT polymer and PIM-1. Consequently, physical aging at longer terms in PIM-1 is moderated by the presence of ABT-based additives.

## 4. Conclusions

Novel hyper cross-linked polymers (HCPs) named ABT01 and ABT02, based on tetraphenyl methane and 1,3,5-tris(bromomethyl)benzene, were successfully synthesized and characterized via FT-IR and ^13^C-SS-NMR spectroscopy, via scanning electron microscopy (SEM) and via N_2_ physisorption analysis at 77 K. By changing the solvent and the temperature of the reaction mixture, control over the particle dimension could be exerted. Dichloroethane and a higher temperature resulted in higher specific surface area and pore volumes.

ABT materials were tested as fillers for the production of PIM-1 based mixed-matrix membranes for CO_2_ and N_2_ gas separation applications. Four MMMs were obtained by adding ABT01 and ABT02 with a 3 and 10 wt % loading with respect to PIM-1. The addition of the fillers causes a reduction in permeability performances with respect to pure PIM-1 towards both CO_2_ and N_2_ at t_0_, right after methanol treatment. In terms of CO_2_/N_2_ selectivity, only PIM1-ABT01-3% and PIM1-ABT02-10% showed higher values with respect to pure PIM-1 while the other samples showed a slight decrease in selectivity.

With aging, all the membranes showed a reduction in gas permeability. However, while pure PIM-1 showed a reduction of 85 and 90% for CO_2_ and N_2_ permeability, respectively, MMMs exhibited a visible slowdown of the aging rate after two months from t_0_. After almost three years of aging, MMMs retained approximately 40% of the initial CO_2_ permeability and approximately 25% of the N_2_ permeability with respect to t_0_ values, hence an increase in CO_2_/N_2_ selectivity was also observed.

^13^C spin-lattice relaxation times (T_1_) allowed monitoring of molecular-level dynamics and degree of flexibility of polymers in membranes. It was found that for the pure PIM-1 membrane and the MMM containing 3% ABT01, T_1_ values increased over time with respect to T_1_ values measured at t0. Longer T_1_ values are a sign of reduction of fractional free volumes within the PIM-1 matrix, due to physical aging. However, for the other MMMs the opposite trend was noticed and shorter T_1_ values were observed with respect to those measured for the freshly cast membranes. This indicates that interactions between the PIM-1 polymer matrix and the ABT fillers provide a way to slow down physical aging for PIM-1-based gas separation membranes.

## Figures and Tables

**Figure 1 membranes-11-00463-f001:**
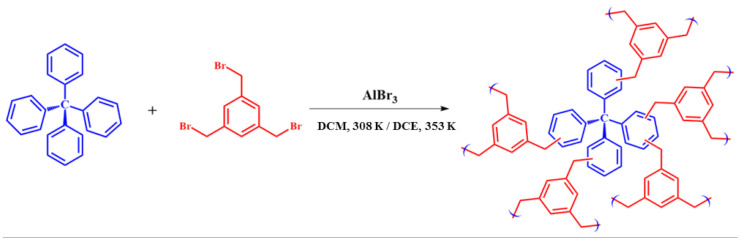
Reaction scheme for the synthesis of ABT hyper crosslinked polymers.

**Figure 2 membranes-11-00463-f002:**
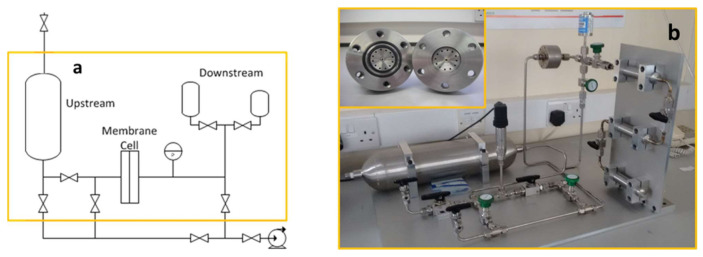
Permeation rig: (**a**) schematic; (**b**) rig with the cell.

**Figure 3 membranes-11-00463-f003:**
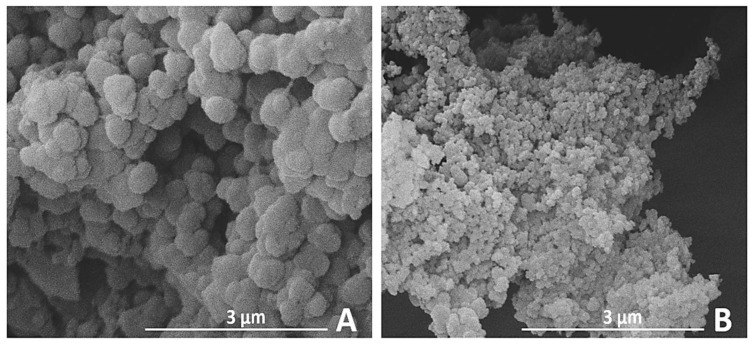
SEM images at 50,000× magnification of ABT01 (**A**) and ABT02 (**B**).

**Figure 4 membranes-11-00463-f004:**
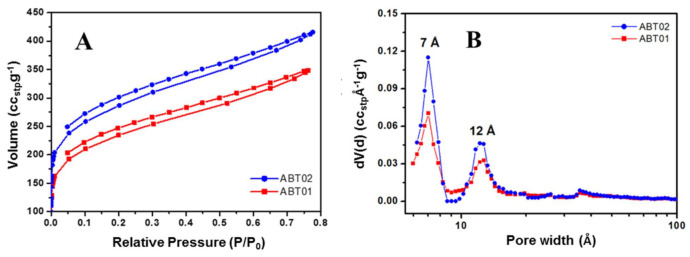
N_2_ physisorption isotherms at 77 K (**A**) and pore size distribution (**B**) of ABT01 (■) and ABT02 (■).

**Figure 5 membranes-11-00463-f005:**
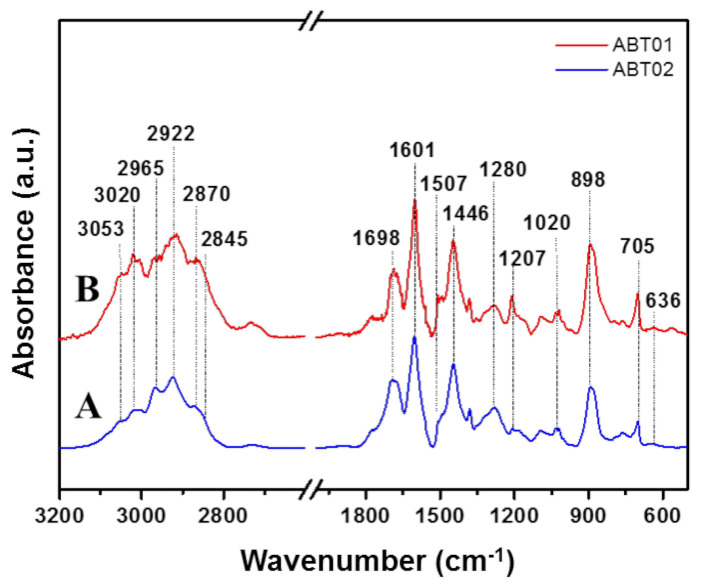
Infrared spectra of ABT01 **―** (**A**) and ABT02 **―** (**B**) acquired under vacuum conditions (minimum pressure below 10^−4^ mbar) and beam temperature. Prior to analysis, the samples were treated at 423 K under vacuum (10^−4^ mbar) for three hours.

**Figure 6 membranes-11-00463-f006:**
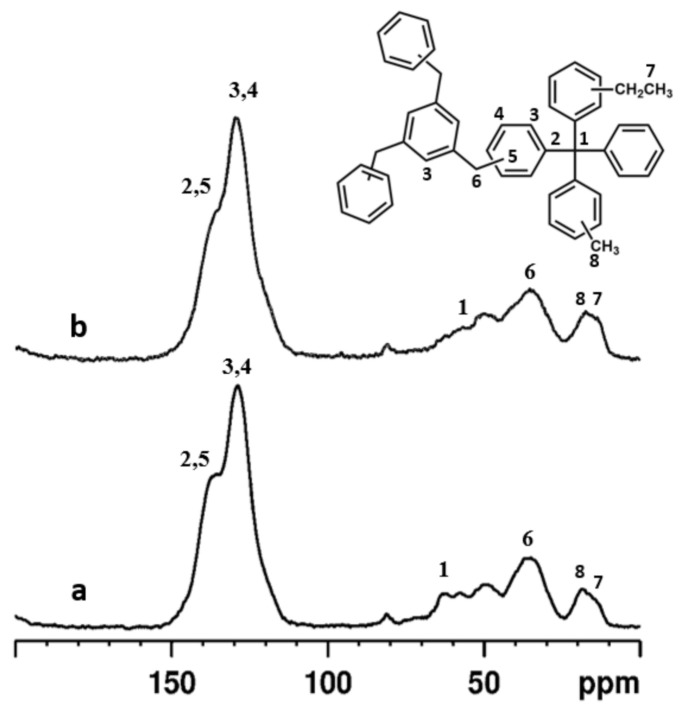
^13^C CPMAS NMR spectra of ABT01 (**a**) and ABT02 (**b**).

**Figure 7 membranes-11-00463-f007:**
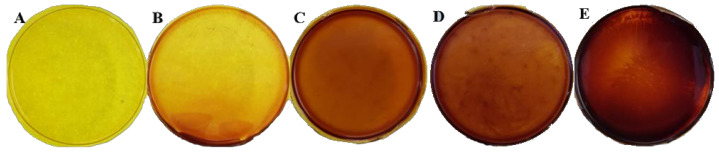
Pristine PIM-1 membrane (**A**), PIM1-ABT01 (3 wt %) (**B**), PIM-ABT02 (3 wt %) (**C**), PIM1-ABT01 (10 wt %) (**D**) and PIM1-ABT02 (10 wt %) (**E**).

**Figure 8 membranes-11-00463-f008:**
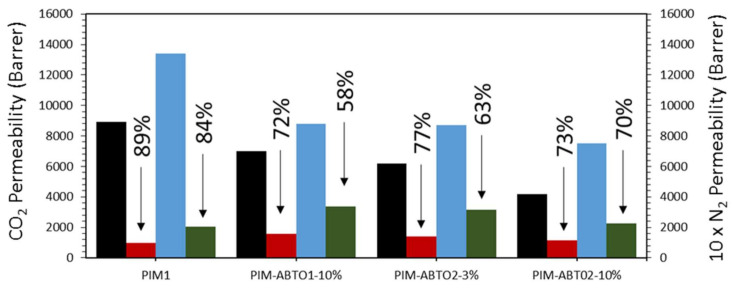
Permeability values and loss of permeability in % of mixed-matrix membranes (MMMs). (

) N_2_ permeability on t_0_; (

) N_2_ permeability on t_f_; (

) CO_2_ permeability on t_0_; (

) CO_2_ permeability on t_f_ (850 days).

**Figure 9 membranes-11-00463-f009:**
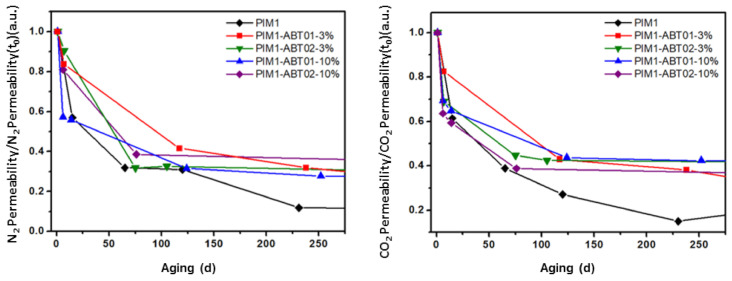
Normalized permeability data with respect to t_0_ values of PIM-1 (◆), PIM1-ABT01-3% (■), PIM1-ABT01-10% (▲), PIM1-ABT02-3% (▼), PIM1-ABT02-10% (◆), as function of time over 1 year. Lines are drawn to guide eye.

**Figure 10 membranes-11-00463-f010:**
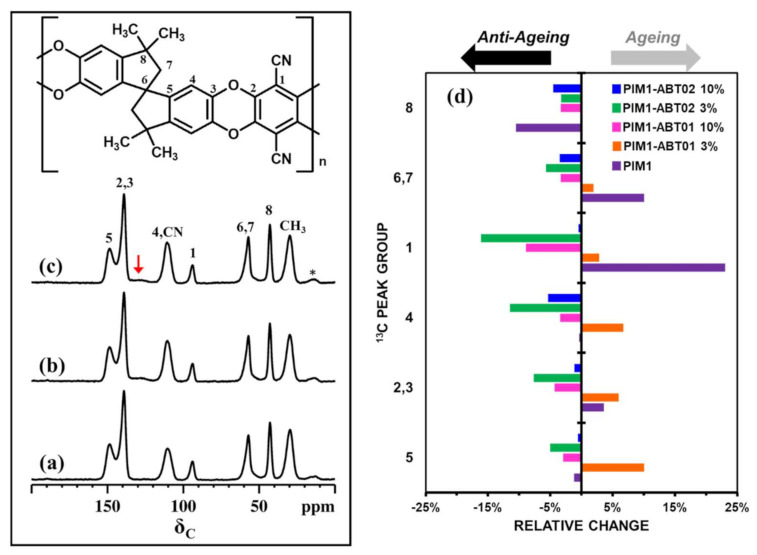
^13^C CPMAS NMR spectra of various PIM-1 based membranes: PIM-1 (**a**), PIM1-ABT01 10% (**b**) and PIM1-ABT02 10% (**c**). The spectra were recorded using a magic angle spinning (MAS) rate of 12 kHz and a CP contact time of 10 ms. * Denotes spinning sidebands. Bar chart (**d**) representing percentage changes in T_1_ relaxation times of chemical groups assigned in (**c**) between t_12_ (1-year aging) and t_0_ (MeOH treatment).

**Table 1 membranes-11-00463-t001:** Textural properties of ABT materials, assessed via N_2_ physisorption analysis performed at 77 K.

Sample	SSA_BET_ (m^2^/g)	V_Tot_ (cc/g)	V_micro_ (cc/g)			V_meso_(cc/g) 20 < Å < 100
			Total	<7Å	7 < Å < 20	
ABT01	823	0.52	0.28	0.08	0.20	0.24
ABT02	990	0.61	0.35	0.10	0.25	0.26

**Table 2 membranes-11-00463-t002:** Assignments of the main IR absorption bands of ABT materials.

Band Positions (cm^−1^)	Assignments [39,40]
3053	ν_As_ Aromatic C-H
3020	ν_S_ Aromatic C-H
2965	ν_As_ Aliphatic C-H (-CH_3_)
2922	ν_As_ Aliphatic C-H (-CH_2_-)
2870	ν_S_ Aliphatic C-H (-CH_3_)
2845	ν_S_ Aliphatic C-H (-CH_2_-)
1280	ν skeletal -C-C-
1700–1210	Collective stretching vibrations of poly-substituted benzene rings
900–700	Collective bending vibrations of poly-substituted benzene rings
636	ν aliphatic C-Br (–CH_2_Br)

**Table 3 membranes-11-00463-t003:** Permeation data for PIM-1, PIM-ABT01 (3 and 10% wt), PIM-ABT02 (3% and 10% wt) at t_0_ (MeOH treatment). (1 Barrer = 10^−10^ cm^3^ (STP)·cm·cm^−2^·s^−1^·cmHg^−1^).

Filler	% wt	Permeability CO_2_ (Barrer) (±5%)	Selectivity CO_2_/N_2_
ABT01	0	13,400	15
	3	14,700	18
	10	8800	13
ABT02	0	13,400	15
	3	8690	14
	10	7500	18

**Table 4 membranes-11-00463-t004:** CO_2_ and N_2_ permeability coefficients and selectivity data of PIM-1, PIM1-ABT01-3%, PIM1-ABT01-10%, PIM1-ABT02-3%, PIM1-ABT02-10% at t_0_ and t_f_.

Membrane	P_CO2_ (Barrer)		P_N2_ (Barrer)		Selectivity CO_2_/N_2_	
	t_0_	t_f_	t_0_	t_f_	t_0_	t_f_
PIM-1	13,400	2040	890	100	15	21
PIM1-ABT01-10%	8800	3390	700	160	13	21
PIM1-ABT02-3%	8690	3170	620	140	14	22
PIM1-ABT02-10%	7500	2270	420	110	18	20

## Data Availability

The data presented in this study are available on request from the corresponding author. The data are not publicly available due to privacy.

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
