# Peer review of "Hyper Cross-Linked Polymers as Additives for Preventing Aging of PIM-1 Membranes"

_membranes, 2021, doi:10.3390/membranes11070463_

Round 1

Reviewer 1 Report

In the work, hyper crosslinked polymers as fillers were applied to prevent aging of PIM-1 membranes. My comments are as follow.

  1. 1 showed such indistinct SEM images, and the quality must be improved.
  2. In Fig.9, why the tests were carried in the same aging days? And for PIM-1, N2 permeability wasn’t tested on ~230th day?
  3. Please explain the mechanism of enhancing anti-aging performance, such as good compatibility, some references could be supplemented, such as “Journal of Membrane Science 591 (2019) 117343”, “Journal of Membrane Science 608 (2020) 118173” and “AIChE J. 66 (2020) e16543"
  4. The “Conclusion” section needs to be refined.

Author Response

We would like to thank the Reviewer for their thoughtful comments, which we feel have improved the manuscript.

Please find attached point-by-point response to Reviewer’s concerns. We hope that you will find our responses satisfactory.

Reviewer 2 Report

-Title: Please avoid using abbreviations (i.e. PMI1) in the title for more clarity. In the case of usage, it must be defined, preferably, immediately in the abstract. Please see “Membranes 20199(3), 41; https://doi.org/10.3390/membranes9030041

-Title: Please remove the dot at the end of the title

-Abstract: The sentence “Then, MMMs based on these 2 HCPs and PIM1 have been developed with 3% and 10%wt loading and…” is not clear. Did you use MMMs and PIM1 for the same meaning? Did you just change the %wt of HCPs? Please rewrite it to make it clear.

-Abstract: Polymers are defiantly synthesized within a size range please provide them 500±?nm and 100±?nm.  

-Introduction, Line 56-57: “The formation of highly interconnected network results into a rigid and stable nanoporous structure”. Based on my experience in molecularly imprinted polymers, a highly crosslinked polymer, the highly interconnected network is stable but could not provide a rigid nanoporous structure as they swell and shrink easily depending on their surrounding environment, e.g. humidity. Please provide an explanation for your sentence with suitable references.

Besides, you explained in the abstract that “ Mixed matrix membranes (MMMs) are membranes that are composed of polymers em-13 bedded with inorganic particles. By combining the polymers with the inorganic fillers, improve-14 ments can be made to the permeability compared to the pure polymer membranes due to new path-15 ways for gas transport”. Inorganic fillers are rigid and can help to modify the permeability of the membrane. How could a polymer with a shrinking/swelling effect help to provide a stable permeability, independent of the surrounding environment?

- Materials and Methods: For the synthesis of ABT01 and ABT02, solvent and temperature are changed. However, the size of the product is related to the type of solvent. Does temperature affect the sizes?

-Line 233: please correct the “Figure 3”.

-Lines 235-236: “In particular, a distinction 235 between type I and II is difficult.” Is it? Is the difference between 990 and 823 m2/g small for your purposes?

-Lines 270-272: Generally, IR methods are not suitable for quantitative purposes. Can you use two FTIR spectrums and compare their intensities to conclude that one is more cross-linked. If yes, please provide a reference.

Author Response

(The authors gave the same response as above.)

Reviewer 3 Report

The paper is quite good. Some methods, like NMR, can be omitted. In my opinion, there is no interesting information in this field.

I would like to ask you to give more information about the method of permeation measurements. It is not enough to send the readers to your former paper. What pressures did you use? How do you obtain permeability in time lag apparatus?...

The English language seems to be good, but some sentences are not understandable. Please, read your paper carefully once more.

Author Response

(The authors gave the same response as above.)

Round 2

Reviewer 2 Report

-Abstract: Polymers are defiantly synthesized within a size range please provide them 500±?nm and 100±?nm. The errors related to the size particles have been added in the Results part. The values are: 498± 39 nm for ABT01 and 120 ± 23 nm for ABT02.

Comment: Please add the exact data that you obtained throughout the manuscript whether in the abstract or in the result sections.

-Introduction, Line 56-57: “The formation of highly interconnected network results into a rigid and stable nanoporous structure”. Based on my experience in molecularly imprinted polymers, a highly crosslinked polymer, the highly interconnected network is stable but could not provide a rigid nanoporous structure as they swell and shrink easily depending on their surrounding environment, e.g. humidity. Please provide an explanation for your sentence with suitable references. The term “rigid” is coming from the reference “Hypercrosslinked porous polymer materials: design, synthesis, and applications. Chem. Soc. Rev., 2017, 46, 3322” and is referred to the aromatic nature of the polymeric framework of the HCPs. In order to avoid confusions, we removed the term “rigid” in the sentence.

Comment: The authors did not answer my question. Did your synthesized polymer swell and shrink depending on the change of environment e.g. shrink when they are dried and swell in contact with moisture? when yes, how did they provide a stable porosity within the membrane?

Besides, you explained in the abstract that “ Mixed matrix membranes (MMMs) are membranes that are composed of polymers embedded with inorganic particles. By combining the polymers with the inorganic fillers, improvements can be made to the permeability compared to the pure polymer membranes due to new pathways for gas transport”. Inorganic fillers are rigid and can help to modify the permeability of the membrane. How could a polymer with a shrinking/swelling effect help to provide a stable permeability, independent of the surrounding environment? The presence of favourable interactions i.e. non-covalent interactions, help to fix the PIM1 polymeric chains in place thus reducing further movements over time which are associated with physical aging of the PIM1 matrix.

Comment: Please see my previous comment. The question is not answered. Please provide suitable references for your claims.
